# Allosteric Modulation of GPCRs of Class A by Cholesterol

**DOI:** 10.3390/ijms22041953

**Published:** 2021-02-16

**Authors:** Jan Jakubík, Esam E. El-Fakahany

**Affiliations:** 1Department of Neurochemistry, Institute of Physiology Czech Academy of Sciences, 142 20 Prague, Czech Republic; 2Department of Experimental and Clinical Pharmacology, University of Minnesota College of Pharmacy, Minneapolis, MN 55455, USA

**Keywords:** GPCRs, cholesterol, allosteric modulation

## Abstract

G-protein coupled receptors (GPCRs) are membrane proteins that convey extracellular signals to the cellular milieu. They represent a target for more than 30% of currently marketed drugs. Here we review the effects of membrane cholesterol on the function of GPCRs of Class A. We review both the specific effects of cholesterol mediated via its direct high-affinity binding to the receptor and non-specific effects mediated by cholesterol-induced changes in the properties of the membrane. Cholesterol binds to many GPCRs at both canonical and non-canonical binding sites. It allosterically affects ligand binding to and activation of GPCRs. Additionally, it changes the oligomerization state of GPCRs. In this review, we consider a perspective of the potential for the development of new therapies that are targeted at manipulating the level of membrane cholesterol or modulating cholesterol binding sites on to GPCRs.

## 1. Introduction

G-protein coupled receptors (GPCRs) are membrane proteins that pass on extracellular signals to the cell using heterotrimeric GTP-binding proteins (G-proteins). GPCRs are integral membrane proteins that possess seven transmembrane α-helices (denoted TM1 to TM7) connected with three intracellular (IL1 to IL3) and three extracellular (EL1 to EL3) loops (Figure 1A). The cysteine in the middle of ECL2 forms disulfide bridge with cysteine at the edge of TM3. The N-terminus of GPCR is oriented out of the cell and may be glycosylated at asparagine or glutamine residues. The C-terminus is oriented to the cytoplasm and may be palmitoylated or myristoylated at cysteine residues. Individual amphiphilic TMs form a circular bundle with a hydrophilic pocket among them that is accessible from the extracellular side (Figure 1B). The pocket serves as an orthosteric site for endogenous transmitter or hormone. More than 30% of currently marketed drugs act at GPCRs and thus GPCRs represent a very important pharmacological target [1].

Most of neurotransmitters act at several receptor subtypes of a given receptor family. This divergence allows one signalling molecule to elicit different cellular responses depending on the distribution of the receptor subtypes in the body. For a pharmacological agent to target body organs selectively, it has to be able to differentially influence the activation of individual receptor subtypes. In general, the binding site for a given endogenous signalling molecule is conserved among its receptor subtypes. This is necessary for accommodating the signalling molecule during the evolution of receptor subtypes. The sameness of the orthosteric site, however, makes finding subtype-selective compounds acting at the orthosteric binding site extremely difficult. In contrast to orthosteric sites, secondary allosteric binding sites on receptors are not under such evolutionary pressure and vary among subtypes [2]. Therefore, a lot of effort was given to the research of allosteric binding sites and allosteric modulators. A large number of various allosteric modulators of GPCRs that bind to the extracellular or intracellular domains were identified.

Cholesterol (CLR) is a sterol-like type of lipid. CLR composes about 30% of all animal cell membranes. The primary function of CLR is structural. It regulates membrane fluidity. Other non-structural functions of CLR include its physical interaction with many membrane proteins including GPCRs (Figure 1B). This interaction results in alteration of receptor properties in terms of the processes of ligand binding, receptor activation and signal transduction [3,4]. Thus, membrane CLR can be considered an allosteric modulator of GPCRs possessing its own specific allosteric binding site.

The family of GPCRs is a large group of evolutionarily-related proteins divided into six classes (termed A to F) that substantially differ in their structure. In this review, therefore, we will focus on class A of GPCRs, also known as rhodopsin-like class. Class A GPCRs includes receptors that govern key physiological processes whose malfunction is associated with various pathologies, e.g., the state of activation of serotonin, dopamine, and muscarinic receptors are involved in mood disorders, Parkinson’s disease, and Alzheimer’s disease, respectively. We review the evidence that membrane CLR interacts with specific binding sites on GPCRs and allosterically modulates binding and action of orthosteric ligands, and receptor oligomerization and signalling. We also explore approaching pharmacologic modulation of membrane cholesterol and modulation of CLR-binding sites as potential therapeutic targets.

## 2. Chemical Properties of Membrane CLR

CLR is a polycyclic and amphiphilic molecule that is found in high abundance in cell membranes. Its main function is regulation of membrane fluidity by facilitation of the formation of ordered phases in the lipid bilayer via composite interactions between lipid components. CLR is a shorter and more rigid molecule in comparison with phospholipids. Therefore, parts of the membrane close to CLR molecules are more rigid and thinner. Fluidity and thickness of the membrane, in turn, affect membrane protein trafficking.

CLR has a flat asymmetric structure defined by a planar α-face and rough β-face, named according to the nomenclature of ring compounds [5]. CLR in the membrane may exist as monomer or form dimers oriented α-face to α-face, the so-called face-to-face dimers, stabilized by Van der Waals contacts (Figure 2A) [6]. Face-to-face CLR dimers were found in X-ray crystal structures of membrane proteins. Another type of CLR dimer may be stabilized by a hydrogen bond between hydroxyl groups (Figure 2B). However, CLR hydroxyl group rather interacts via hydrogen bonding with other membrane lipids or proteins [7]. The third type of CLR dimers, trans-bilayer tail-to-tail dimer, has been hypothesized to exist in membranes (Figure 2C) [8].

## 3. General Mechanisms of Cholesterol Action at GPCRs

In principle, CLR may affect GPCRs in two ways. It may either directly bind to the receptor and thus allosterically modulate the affinity of ligands, efficacy of agonists and spontaneous activity of the receptor. Alternatively, CLR may affect GPCRs indirectly by changing fluidity and organization of the membrane that in turn affects signalling of GPCRs. Direct modulation of GPCRs by CLR requires its interaction with specific sites on receptors with sufficient affinity. Such sites were identified in many GPCRs; see below. In contrast, the indirect mechanism does not involve CLR-specific binding site. As stated above, CLR decreases membrane fluidity that slows down the diffusion of solute molecules like receptors, channels or membrane enzymes that in turn slows-down kinetics and decreases the efficacy of signal transmission from a given receptor to its effector. In membranes rich in CLR content, CLR has a propensity to associate into patches. High CLR content increases the order of neighbouring acyl chains that leads to increased bilayer thickness [9]. These membrane microdomains are termed lipid rafts and substantially affect signal transduction [10]. The hydrophobic mismatch is defined as the difference between the hydrophobic membrane thickness and the peripheral length of the hydrophobic part of the membrane-spanning protein [11]. The membrane-perpendicular length of GPCRs is shorter in an inactive conformation than in an active conformation. Therefore, GPCRs in an inactive conformation may be preferentially sorted to non-raft regions that represent a thinner part of the membrane. Consequently, keeping a receptor in the non-raft region may constrain it in an inactive conformation [12]. Thus, keeping a receptor in non-raft region ablates its signalling. Moreover, the differential localization of proteins in various microdomains increases the specificity of signalling. Co-localization of several signalling pathways at a given microdomain, for example, may promote the formation of a signalling hub that enables integration of distinct signalling pathways at the receptor-membrane interface [13,14]. Thus, lipid rafts play a unique role in cell physiology and pathology and represent possible target in hematopoietic, inflammatory, neurodegenerative, and infectious diseases [15]. Taken together, the indirect effects of membrane CLR are diverse and bring complexity to GPCR signalling.

## 4. Binding of Cholesterol to GPCRs

CLR was found co-crystallized with many GPCRs of class A suggesting possible specific binding. At the time of writing of this review, 44 X-ray or cryo-EM structures of 18 receptors of GPCRs of Class A have been published in the RCSB database (https://www.rcsb.org/, accessed on 20 January 2021) (Table 1). CLR was found co-crystallized with receptors for structurally different agonists including biogenic amines like adrenaline (α_2C_, β_2_) or serotonin (5-HT_2B_), peptides like angiotensin (AT_1_), chemokines (CCR9, CXCR2, CXCR3), endomorphins (κOR, μOR), endothelin (ETB), formyl peptide (FPR2) or oxytocin (OTR), purines like adenosine (A_2A_) or ADP (P2Y_1_, P2Y_12_), endocannabinoids (CB_1_, CB_2_), and eicosanoids like leukotriene (CLT2). CLR in crystals appeared as monomer or dimer. For some receptors, CLR binding was confirmed in several crystal structures, e.g., β_2_, 5-HT_2B_, or A_2A_. On the other hand, for some receptors, X-ray structures provide contradictory results, e.g., AT_1_, CXCR2 or ETB. It also should be noted that in some cases CLR was found in an unexpected orientation, for example, parallel to membrane (CB_2_) or hydroxy group in the middle of membrane bilayer (CB_1_). Additionally, no CLR was found in the crystal structures of GPCRs at which CLR was shown to have a profound effect on ligand binding or receptor activation; see below. Thus, information on the interaction of CLR with GPCRs inferred from crystal structures should be taken with caution. Further, cholesteryl hemisuccinate (CHS) that is used for solubilisation of biological membranes was found co-crystallized with GPCRs. CHS may compete out CLR from binding to GPCR. As it is not certain whether co-crystallized CHS indeed binds to the CLR-specific binding site on GPCRs or is the result of the solubilization process, CHS binding to GPCRs is not covered in this review.

Several CLR sites can be distinguished (Table 1, Figure 3). A CLR dimer in the outer leaflet of membrane binds to A_2A_-adenosine receptor at a groove between TM2, TM3, and TM4 (Figure 3A) or TM6 (Figure 3C). A similar binding of CLR as in Figure 2A can be found at the P2Y_1_ receptor (4XNV) (Table 1). The κ-opioid receptor in an active conformation (6PT2, 6B73), the μ-opioid receptor at an active (5C1M) or inactive conformation (4DKL) and the CXCR3 receptor at an active conformation (5WB2) have CLR bound to the same site as the A_2A_-adenosine receptor shown in Figure 3C. A CLR monomer in the outer leaflet binds to the oxytocin receptor at TM4 and TM5 (Figure 3B) or to α_2C_-adrenergic receptor at TM1 and TM7 (Figure 3D). A similar binding site as in Figure 3D has been identified at the endothelin receptor (5X93) and purinergic P2Y_12_ receptor (4NTK) (Table 1). A CLR dimer in the inner leaflet of the membrane binds to the β_2_-adrenergic receptor in a groove formed by TM2, TM3, and TM4 (Figure 3E). The same CLR-binding site is present at the CXC receptor 2 (6LFM), formyl peptide receptor 2 (6CW5, 6OMM) and P2Y_12_ purinergic receptor (4NTJ). A CLR monomer in the inner leaflet binds to TM6 of an inactive conformation of the κ-opioid receptor (Figure 3F) or the CCR9 chemokine receptor (Figure 3G). A similar binding of CLR as in Figure 3G has been found at the cysteinyl leukotriene receptor 2 (6RZ7) and formyl peptide receptor 2 (6CW5, 6OMM). A special kind of CLR interaction at TM1 stabilized by palmitic acid covalently bound to cysteine in helix 8 can be found at the 5-HT_2B_ receptor (Figure 3H) and also at the β_2_-adrenergic receptor (2RH1) and angiotensin receptor 1 (6OS1).

Receptors found co-crystallized with CLR mediate their primary functional responses via all three major subclasses of G-proteins: G_i_, G_s_, and G_q_. None of the CLR-binding sites can be considered typical for a given receptor coupling pathway, suggesting that CLR-binding sites evolved independently from receptor coupling. Similarly, comparison of GPCRs in active and inactive conformations does not show any correlation with CLR binding. This suggests the absence of a common mechanism of CLR action on receptor activation.

Based on X-ray structures two putative cholesterol-binding motifs were postulated. Besides the so-called ’CLR recognition amino acid consensus’ (CRAC) domain common for all membrane proteins [41], the so-called ‘CLR Consensus Motif’ (CCM) was identified in the structure of the β_2_-adrenergic receptor (3D4S) [17]. CCM is the groove formed by 2 or 3 TMs. For the 3D4S structure, residues R151, L155 W158 in the TM4 and Y70 in the TM2 were identified as key CLR-binding residues (Figure 4). Although the orientation of the CLR dimer in the 2RH1 structure is slightly different from the 3D4S structure, key interactions with CCM are preserved [16]. The same applies to binding of the monomeric CLR in the 6PS0 structure [18]. In contrast, no CLR was found in four structures of the β_2_-receptor: 2R4R, 2R4S, 3KJ6, 3P0G. Based on bioinformatics studies of GPCR homology, the consensus sequence of CCM has been established as R/K-X_5_-I/V/L-X_5_-Y/W in the one helix and F/Y in the opposing helix. Residues R/K and F/Y of CCM are at the intracellular edge of TM helices. Residues Y/W are approximately in the middle of the membrane. The hydroxyl group of CLR interacts with a basic residue of CCM that are abundant at the intracellular edge of TMs. The β-face of the CLR dimer binds strongly with W or Y via hydrophobic, mainly π-π stacking, interactions. The CRAC domain (R/K-X_5_-Y-X_5_-L/V) and its reversed CARC (L/V-X_5_-Y-X_5_-R/K) are similar to the CCM in having R or K at the edge of the membrane. In comparison to CCM, positions of aromatic and hydrophobic residues are swapped in CRAC and CARC. While CRAC and CARC accommodate monomeric CLR, the CCM may bind a CLR dimer.

Although CCM, CRAC, and CARC motifs appear in the sequence of large number GPCRs [17,42], CLR was found in structures lacking a CLR-binding motif, e.g., cannabinoid CB_2_ receptor (6PT0) or endothelin receptor (5X93) (Table 2). Binding of CLR to a detected CLR-binding motif was confirmed only in some of the published structures. For example, CCM was found at all five subtypes of the muscarinic acetylcholine receptor. However, no CLR was detected at any of the 16 published structures. In structures of the M_1_ receptor (5CXV and 6WJC), CHS is bound to CCM [43,44]. In many structures possessing a CLR-binding motif, CLR-binding was detected somewhere else. One of the abundant non-canonical CLR-binding sites is in the inner leaflet of the membrane at TM1 and Helix 8, e.g., structures 2RH1 (β_2_-adrenergic); 4IB4, 5TVN, and 6DRX (5-HT_2B_); and 6OS1 (AT_1_). The binding of CLR in this site is stabilized by palmitic acid covalently bound to the cysteine in Helix 8. Another non-canonical CLR-binding site appears in the outer leaflet of the membrane at TM6, e.g., structures 6PT2 (δ-opioid); 6B73 (κ-opioid); 4EIY and 5IU4 (A_2A_-adenosine). At structures 4DLK and 5C1M of the μ-opioid receptor, CLR binds to the variation of CCM. In these structures, the CLR hydroxyl group makes a hydrogen bond with Q314 instead of basic R or K of classic CCM. At structure 6LFM of the CXCR2 receptor, CLR binds to the variation of CRAC that possesses W instead of Y.

Besides X-ray crystallography, approaches of computational chemistry also predicted interaction of CLR with GPCRs at many sites [4,45,46]. Multi-scale simulations of molecular dynamics revealed that CLR-interaction sites are dynamic in nature and are indicative of ‘high occupancy sites’ rather than ‘binding sites’. The results suggest that the energy landscape of CLR association with GPCRs corresponds to a series of shallow minima separated by low barriers. However, extensive all-atom simulations of molecular dynamics of the β_2_-adrenergic receptor (3D4S) suggest that CLR interacts specifically with the CCM and its binding is stable over the course of simulation [17]. CLR binding to the CCM of the β_2_-adrenergic receptor requires a slow, concerted rearrangement of side chains [47].

The structures of the A_2A_ adenosine receptor 4EIY and 5UI4 contain three and four CLR molecules, respectively, that are bound at two nearly opposite positions at the extracellular side of the receptor (Figure 3A,C) [26,27]. However, none of them interacts with CCM detected at the intracellular half of TM2 and TM4. Simulation of molecular dynamics of the system containing the A_2A_-adenosine receptor in lipid bilayer containing 30 % of CLR resulted in the association of CLR with CCM and stabilization of TM6 by the CLR dimer [26,48]. The same approach identified additional CLR binding sites on the A_2A_ receptor [49].

Two molecules of CLR were successfully docked to the site at the intracellular half of TM6 of the M_1_ muscarinic receptor (5CXV) identified by site-directed mutagenesis (Figure 5) [50]. Simulation of molecular dynamics of the docked CLR confirmed the stability of CLR binding and identified the hydrogen bond to R365 (R6.35 according to Ballesteros-Weinstein numbering [51]) as the key interaction.

## 5. Effects of CLR on Ligand Binding

Ligand binding to GPCRs can be modulated by CLR in two ways: (i) CLR alters membrane fluidity that in turn affects the conformation of the receptor and its affinity for a given ligand or (ii) CLR specifically binds to the receptor and allosterically changes ligand affinity. From a pharmacological point of view, CLR specific binding and allosteric receptor modulation are more relevant than CLR effects on membrane fluidity, as they offer the possibility of selective modulation of individual GPCRs.

Thermostability and NMR studies of the β_2_-adrenergic receptor suggested specific CLR binding with affinity as high as 1 nM [52]. CLR bound to GPCR may modulate it at several levels. It may affect ligand binding, receptor function, or receptor oligomerization. The human variant of the oxytocin receptor (OTR) expressed in Sf9 cells that are naturally lean in CLR has low affinity for oxytocin. Oxytocin high-affinity binding appears and increases with an increase in the content of membrane CLR [53]. Thus, in the case of OTR, oxytocin exerts positive cooperativity with CLR. According to the crystal structure, membrane CLR binds at the outer leaflet of the membrane at TM4 and 5 (Table 1) that is in the vicinity of the orthosteric site. This may explain the profound effects of CLR on oxytocin binding.

Solubilisation of the hippocampal 5-HT_1A_ receptor by CHAPS that is accompanied by a loss of membrane CLR results in a reduction in specific agonist binding and extent of G-protein coupling. Replenishment of solubilized membranes with CLR enhances specific binding of the agonists and receptor G-protein coupling [54]. Thus, similarly to OTR, CLR exerts positive cooperativity with tested agonists at the 5-HT_1A_ receptor. Further studies have shown that only one of two enantiomers of CLR, *ent*-cholesterol, supports the function of the 5-HT_1A_ receptor like membrane CLR [55].

Membrane CLR impairs chemokine binding to CCR5 receptors [56] but increases chemokine binding to CXCR4 receptors [57]. No CLR was found co-crystallized with CCR5 or CXCR4. Elevated brain cholesterol impairs the affinity of cannabinoids for CB_1_ receptor [58]. CLR was found to bind in the inner leaflet of the membrane to TM3 and TM4 of the CB_1_ receptor in an active conformation. The location of the CLR-binding site on the CB_1_ receptor is close to the activation switch, suggesting that CLR may impair the affinity of CB_1_ agonists indirectly by modulation of receptor activation.

Two CLR dimers are identified in the crystal structure of the A_2A_-adenosine receptor (Table 1). One of them binds in the outer leaflet of the membrane at TM6 that is in a vicinity of the orthosteric binding site. In contrast to OTR, the depletion of membrane cholesterol did not affect ligand binding to the A_2A_-adenosine receptor [48]. In contrast, binding of the A_2A_-adenosine receptor antagonist [^3^H]ZM241385 was partially decreased by 3 mM water-soluble CLR [59]. Based on replica exchange molecular dynamics, the authors suggest that water-soluble CLR gains access to the orthosteric binding site and decreases antagonist binding competitively.

Modulation of ligand binding to muscarinic receptors by CLR varies among subtypes. CLR was not found at any of 16 crystal structures published so far. A possible CLR-binding site was predicted in the inner leaflet of the membrane at TM6 using site-directed mutagenesis [50]. This site seems to have a greater impact on the activation of muscarinic receptors than ligand binding. CLR depletion lowered the affinity of the antagonist *N*-methylscopolamine (NMS) to M_1_, M_2_, and M_3_ subtypes [60,61]. Enrichment of membranes with CLR led to an increase in affinity for NMS at M_2_ but decrease at M_1_ and M_3_ receptors. The effects of CLR on affinity for the agonist carbachol were opposite to those on the affinity of NMS. The profound effects of CLR on ligand binding may be mediated by a CLR-binding site different from the one identified at TM6.

In contrast to κ- and µ-opioid receptors, no CLR was found co-crystallized with δ-opioid receptors. At δ-opioid receptors, effects of CLR on ligand binding are rather mediated by alteration of receptor oligomerization state, see below. Thus, the final effect of CLR on ligand binding differs in CLR-rich and CLR-lean membranes [62].

Taken together the effects of CLR on ligand binding to GPCRs cannot be generalized as there is no correlation between CLR effects and location or structure (binding motif) of the CLR-binding site on the receptor. This variability provides a chance for the development of selective allosteric modulators based on the CLR scaffold targeting the CLR-binding site at the receptor of interest.

## 6. Effects of CLR on the Functional Response of GPCRs

GPCRs are highly dynamic membrane proteins adopting various ligand-specific conformations [63]. NMR of the β_2_-adrenergic receptor revealed that an agonist alone was not able to stabilize an active receptor conformation [64]. Thus, allosteric ligands are expected to profoundly affect the functional response of GPCRs to agonists. CLR may serve as such allosteric modulator. The functional response of GPCRs to their respective agonists may be modulated by CLR in two ways: (i) CLR affects membrane fluidity and in turn, allosteric changes in receptor structure leading to receptor activation or (ii) CLR specifically binds to the receptor and allosterically changes agonist affinity and operational efficacy [65]. Two pools of CLR were identified in simulations of molecular dynamics (i) an outer (annular) shell of CLR rapidly associating with and dissociating from the receptor, and an inner pool of tightly bound (non-annular) CLR molecules [66]. In general, both CLR pools may restrict the ability of the receptor to attain certain conformations. From a pharmacological point of view, both mechanisms are important for the functional response of the receptor. This is in contrast to the case of ligand binding, where specific high-affinity binding of CLR is more important than the effects of CLR on membrane properties.

General mechanism of GPCR activation by agonists starts by binding of an agonist to the receptor in an inactive conformation. Agonists binding elicits changes in molecular switches (transmission switch in TM6 and tyrosine toggle switch in TM7) that propagate change in conformation from the orthosteric binding site to the ionic lock switch at the intracellular edge of TM3 [67]. Agonist induced changes in the ionic-lock switch lead to disruption of the ionic lock between R3.50 and D/E3.49 in the TM3, or in some cases E6.30 in the TM6, that prevents interaction of R3.50 with cysteine in the C-terminus of the α-subunit of G-protein [68]. The rearrangement of molecular switches allows for the relative movement of TM3 and TM6. Due to the proline kink in the middle of TM6 (P6.50), its rotation leads to increase in the distance between intracellular edges of TM3 and TM6 and opening receptor G-protein interface for insertion of the C-terminus of the G-protein α-subunit [69]. Findings of CLR co-crystallized close to R3.50 (e.g., 2RH1, 3D4S), E.30 (e.g., 5LWE), or transmission switch (e.g., 4EIY) suggest that CLR may specifically modulate receptor activation 1.

The observed effects of CLR on receptor activation vary among receptors. Comparison of the effects of various sterols on membrane fluidity (assessed by fluorescence anisotropy) and on the functional response of OTR to oxytocin and CCKR to CCK8 has shown that CLR modulates these receptors employing both possible mechanisms of CLR action [70]. Depletion of membrane cholesterol also attenuated signalling at the 5-HT_1A_ and A_2A_ receptors [48]. Similarly, removal of membrane cholesterol reduced cAMP signalling of the μ-opioid receptor [71]. However, removal of membrane CLR did not affect signalling of the δ-opioid receptor.

A possibility of allosteric modulation of protein function by CLR hydroxy group or even direct interaction with the ionic lock stabilizing the active state of the β_2_-adrenergic receptor was postulated [72]. However, in contrast to OTR, the A_2A_-adenosine, 5-HT_1A_ and μ-opioid receptor, CLR attenuates signalling of the β_2_-adrenergic receptor mainly by separation the receptor from its signalling partners [73,74]. In accordance, extensive atomistic simulations of molecular dynamics of 3D4S structure revealed that CLR makes the β_2_-adrenergic receptor less flexible so that it can only adopt certain conformations [75]. In these simulations, the effects of CLR on the β_2_-adrenergic receptor activation were due to direct binding of CLR to the receptor.

The effects of membrane CLR on signalling vary among muscarinic acetylcholine receptor subtypes. At M_2_ muscarinic receptors, CLR depletion led to an increase in preferential signalling (G_i_-mediated inhibition of cAMP synthesis) as well as non-preferential signalling (G_s_-mediated activation of cAMP synthesis) [60]. Enrichment of the membranes with CLR led to a slight attenuation of both preferential and non-preferential signalling. Effects of high CLR content on non-preferential signalling were more eminent after inactivation of G_i_ G-proteins by pertussis toxin. At M_1_ and M_3_ receptors, CLR had similar effects on non-preferential signalling (G_s_-mediated activation of cAMP synthesis), suggesting that an increase in membrane fluidity facilitates signalling via adenylate cyclase [61]. In contrast, increase as well as a decrease in membrane CLR attenuated preferential signalling (G_q_-mediated stimulation of IP_X_ synthesis) at M_1_ and M_3_ receptors. Besides the effects of membrane fluidity on the signalling of muscarinic receptors, CLR was shown to specifically bind to muscarinic receptors and modulate their activation [50]. Site-directed mutagenesis revealed that CLR binds to the site at TM6 (R/Q6.35 and L/I6.46). From this site, CLR prevents persistent activation of the M_5_ receptor by wash-resistant xanomeline.

## 7. Effects of CLR on Oligomerization of GPCRs

In contrast to class C, GPCRs of class A are in the vast majority of cases functional as monomers [76]. The opioid receptors are among the most studied class A GPCRs in term of oligomerization. In their pioneering study, Jordan and Devi have shown that κ- and δ-opioid receptors heterodimerize to form functionally and pharmacologically diverse receptors [77]. Further studies have shown a key role of µ-δ-opioid receptor heteromers in analgesia and neuropathic pain [78,79]. Oligomerization of purified μ-opioid receptors was confirmed by fluorescent techniques [80]. However, the homo-dimerization of the µ-opioid receptor under physiological conditions is limited [81]. A recent study has shown that homo-dimerization of µ-opioid receptors can be enhanced by certain agonists suggesting that receptor activation may be modulated by dimerization [82]. Homo-dimers of κ-opioid receptors were confirmed by crystallography [20,23]. Rhodopsin dimers were seen in native membranes by means of atomic force microscopy [83]. However further studies showed that rhodopsin can function effectively as a monomer [84]. It was postulated that rhodopsin dimers may be required for binding of arrestins that form dimers [85]. However, monomeric rhodopsin was found to be sufficient for binding of arrestin dimers [86]. Recently, native rhodopsin dimers were found in nanodiscs using cryo-electron microscopy [87].

So far many of GPCRs are known to exist as oligomers that differ from protomers in ligand binding and function [88]. The role of CLR in oligomerization was proposed for some of them, e.g., cannabinoid receptors [89]. An effect of CLR on oligomerization of GPCRs may be direct (CLR is an integral part of the protomer-to-protomer interface) of indirect (CLR affects the organization of the cell membrane that in turn affects the oligomerization process). Indirect as well as direct evidence exists for both mechanisms. An example of indirect effects of CLR on oligomerization is the chemokine CXCR4 receptor. Homo-dimerization of CXCR4 is conditioned by lipid rafts, as evidenced by depletion of membrane CLR that reduced dimerization of the CXCR4 receptor [90]. An example of direct evidence for CLR mediated dimerization is the recently published X-ray structure of the α_2C_-adrenergic receptor (6KUW, Chen et al., to be published). In this structure, CLR in the outer leaflet of the membrane is part of the dimerization interface (Figure 6). The dimer of the α_2C_-adrenergic receptor is symmetric with interface formed by TM1 and TM7 of each protomer. Each of two CLR molecules intercalates between TM1 and TM7 of one protomer and interacts with Q413 in the EL3 of the other protomer via water bridges.

Oligomerization of β_2_-adrenergic receptors was postulated based on the heterogeneity of agonist binding. Constitutive dimers of β_2_-adrenergic receptors were detected in the membranes of living cells by bioluminescence resonance energy transfer (BRET) [91]. These dimers are functional as agonists increased BRET signal indicating agonist-induced dimerization. Coarse-grained simulations of molecular-dynamics were carried out to analyse the interactions between membrane CLR and the β_2_-adrenergic receptor [92]. Results have shown the direct effects of CLR on receptor dimerization. At membranes with low CLR content, the dimer interface was most often a hetero-interface, formed by TM1 and TM2 of one protomer and TM4 and TM5 of the other. With an increase in CLR content, CLR binding to TM4 increased and prevented the formation of hetero-interface. At membranes with high CLR content, the dimer interface was formed mainly by homo-interface, formed by TM1 and TM2 from both receptors. The crystal structure of β_2_-adrenergic receptor binding a partial inverse agonist (2RH1) indicates a possible symmetric arrangement of dimeric receptors with TM4-TM5 to TM1-helix8 interface [16]. The dimer interface is mediated by ordered lipids consisting of six cholesterol and two palmitic acid molecules per receptor dimer (Figure 7). At each receptor, one CLR dimer is bound to CCM between TM2 and TM4 and monomeric CLR is bound to TM1 and helix 8. Palmitic acid is covalently bound to C341 in helix 8.

Oligomerization of 5-HT_1A_ receptors was intensively studied, however, results are complex and capricious. In live cells, time-resolved fluorescence anisotropy revealed constitutive oligomers of 5-HT_1A_ receptors and have shown that the oligomerization of the 5-HT_1A_ receptors is independent of agonist stimulation but acute depletion of membrane CLR increases the number of oligomers [93]. In contrast, the study from the same group using homo-Förster resonance energy transfer (homo-FRET) and fluorescence lifetime imaging (FLIM) confirmed constitutive oligomers of 5-HT_1A_ receptors but have shown that depletion of membrane CLR and antagonist treatment decrease the population of oligomers and stimulation of 5-HT_1A_ receptors with agonists increase the population of oligomers [94]. These apparent contradictions can be explained on the bases of different techniques being able to detect different sub-population of receptor oligomers. Alternatively, receptor oligomerization (and subsequently CLR effects) is dependent on the density of receptors in the membrane [91,95]. Using FRET between CFP and YFP fused to C-terminus of the receptors and site-directed mutagenesis, TM4 and TM5 were identified as a dimerization interface of the 5-HT_1A_ receptors [96]. Coarse-grained simulations of molecular-dynamics at microsecond scale were carried out to study self-assembly mechanisms of 5-HT_2C_ receptors [97]. In CLR-free membrane, 17 different dimerization interfaces were identified. The strongest dimerization was mediated by TM5-TM4 quasi-symmetric and TM1-TM2-helix 8 symmetric dimerization. In CLR-rich membranes, TM1-TM7-helix 8 interface with bound CLR takes prevalence indicating an important structural role of CLR in dimerization at natural membranes that are rich in CLR.

Oligomers of muscarinic acetylcholine receptors were inferred from immunoprecipitation studies of tagged receptors [98,99] and ligand binding not following binary reaction [100,101,102]. Moreover, it was observed that CLR induces heterogeneity in the binding among various radioligands. The proposed explanation is that CLR promotes cooperativity in the binding among antagonists bound to the oligomeric M_2_ receptor [103]. Further multi-photon fluorescence resonance energy transfer studies suggested that CLR may be a constituent of a dimer-to-dimer interface of the M_3_ receptor tetramer [104].

Another group of GPCRs of Class A at which important role of membrane CLR has been detected are chemokine receptors. Depletion of membrane CLR by methyl-β-cyclodextrin ablated CCR5 signalling [105]. CLR increased chemokine binding to solubilized CXCR4 receptors [57]. CLR affects oligomerization and signal trafficking of chemokine receptors [14]. As chemokine receptors do not possess a CCM, effects of CLR on membrane fluidity and organization into rafts were postulated as underlying mechanisms in oligomerization and promotion of the signalling hubs enable signal trafficking. Organization of chemokine receptors in lipid rafts leads to stabilizing particular receptor conformations that are manifested in changes in chemokine binding. Besides indirect CLR effects on oligomerization of chemokine receptors, CLR binding to dimers of CXCR4 was predicted using molecular dynamics simulations [106]. In molecular modelling studies, CLR changes the pattern of CXCR4 dimerization. While in CLR-free phospholipid bilayers CXCR4 dimerizes via TM1 to TM5-TM7 interface, in the presence of CLR CXCR4 dimerizes through the symmetric TM3-TM4 to TM3-TM4 interface intercalated by cholesterol molecules.

## 8. Perspectives

Evidence for the role of membrane CLR in GPCRs binding, activation, signalling, and oligomerization is overwhelming. Balanced levels of membrane CLR determine the proper function of GPCRs. Moderate fine-tuning of CLR levels thus represents a therapeutic opportunity in conditions with an altered CLR level. Membrane CLR facilitates the formation of microdomains termed lipid rafts. Lipid rafts affect the specificity and efficacy of GPCR signalling and represent another CLR-related target in experimental therapeutic.

High-affinity binding of CLR to a receptor is very common among GPCRs of class A. It was demonstrated not only to canonical sites like cholesterol consensus motif (CCM) and CLR recognition amino acid consensus’ (CRAC/CARC) but also to non-canonical ones. These findings open a way for the development of new high-affinity allosteric modulators of GPCRs based on steroid scaffold of CLR. For example, several neurosteroids that share a steroid scaffold with CLR have been shown to exert their non-genomic effects via muscarinic [107,108], serotonin [109], or α_2_-adrenergic receptors [110]. The variability in the CLR effects and binding motifs, as well as different location of CLR-binding sites among GPCRs, gives a chance for the development of selective allosteric modulators based on sterol structure targeting the CLR-binding site. The possibility to achieve pharmacological selectivity based on receptor-membrane interactions is a completely new approach in pharmacotherapy.

To facilitate the development of CLR-oriented therapies detailed picture of the action of membrane CLR on GPCRs binding, activation, signalling, and oligomerization is needed. To this end, one needs to apply state-of-the-art techniques for analysis of changes in the organization of the membrane and interactions among membrane components. Advanced fluorescent techniques like BRET, FRET-FLIM, and anisotropy-FRET represent promising approaches.

## Figures and Tables

**Figure 1 ijms-22-01953-f001:**
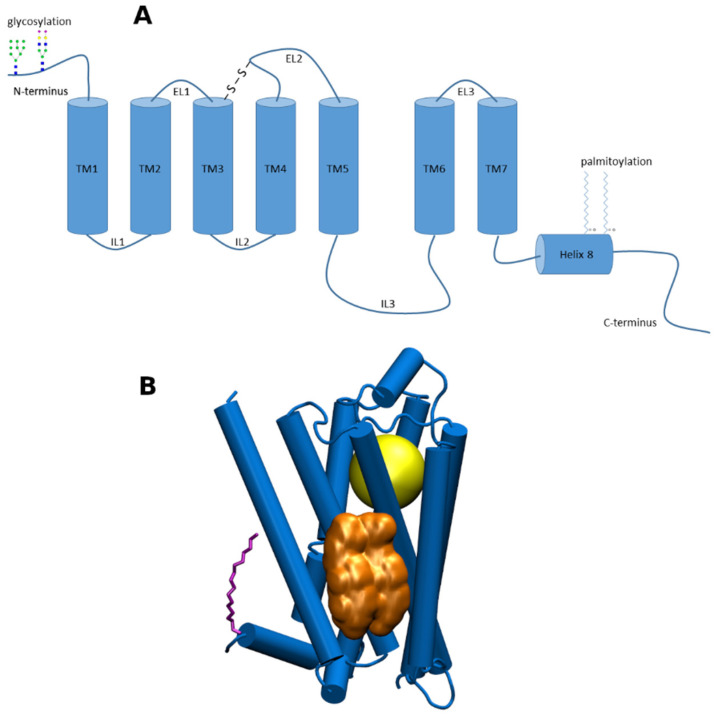
(**A**), a schematic representation of structural features of G-protein coupled receptors (GPCRs) of class A; (**B**) 3D representation of GPCR of class A. Yellow—the orthosteric binding site; gold—cholesterol dimer bound to cholesterol consensus motif (CCM); purple—palmitic acid covalently bound to cysteine in Helix 8.

**Figure 2 ijms-22-01953-f002:**
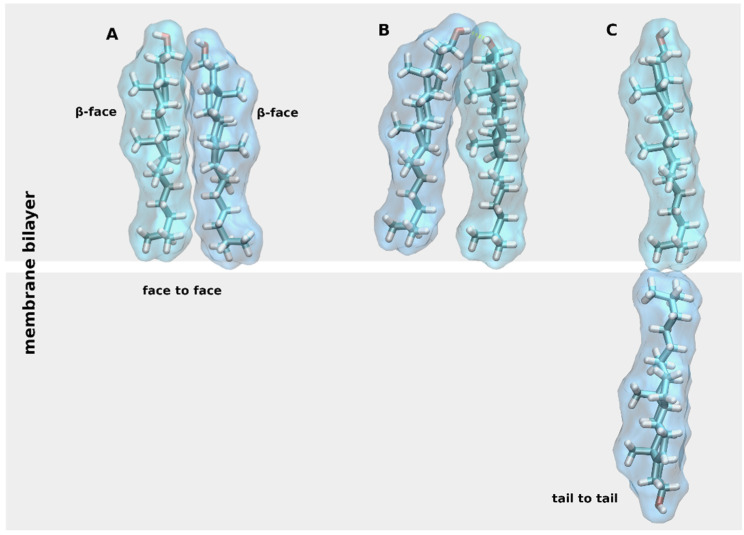
Cholesterol (CLR) dimers. Three types of CLR dimers: (**A**), face-to-face dimer; (**B**), dimer stabilized by a hydrogen bond (yellow dashed line); (**C**) tail-to-tail dimer.

**Figure 3 ijms-22-01953-f003:**
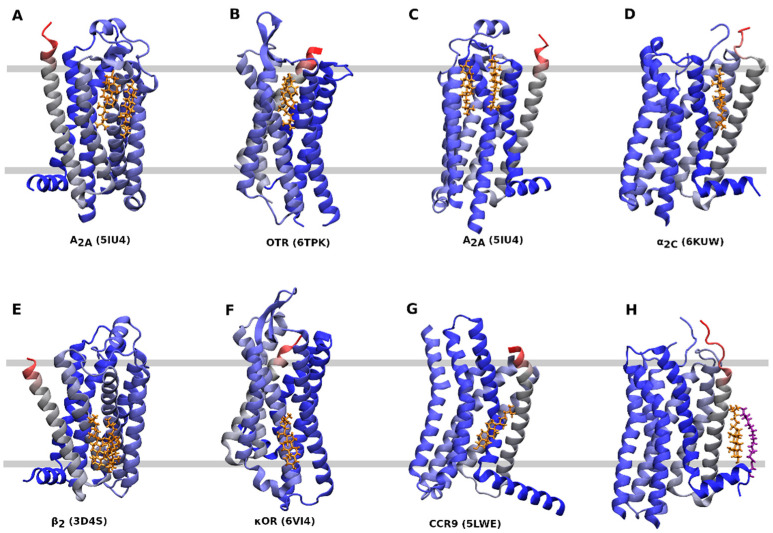
CLR binding sites. Orientation—extracellular side up, N-terminus—red, C-terminus— blue, cholesterol—gold, palmitic acid—purple. (**A**), CLR dimer binding to A_2A_-adenosine receptor (5IU4) at TM2, TM3 and TM4. (**B**), CLR monomer binding to oxytocin receptor (6TPK) at TM4 and TM5. (**C**) CLR dimer binding to the A_2A_-adenosine receptor (5IU4) at TM6. (**D**), CLR monomer binding to α_2C_ adrenergic receptor (6KUW) at TM1 and TM7. (**E**) CLR dimer binding to the β_2_ adrenergic receptor (3D4S) at TM2, TM3, and TM4. (**F**), CLR monomer binding to the κ-opioid receptor (6VI4) at TM4 and TM5. (**G**), CLR monomer binding to the CCR9 chemokine receptor (5LWE) at TM6. (**H**), CLR monomer binding to the 5-HT_2B_ receptor (4IB4) at TM1 and helix 8.

**Figure 4 ijms-22-01953-f004:**
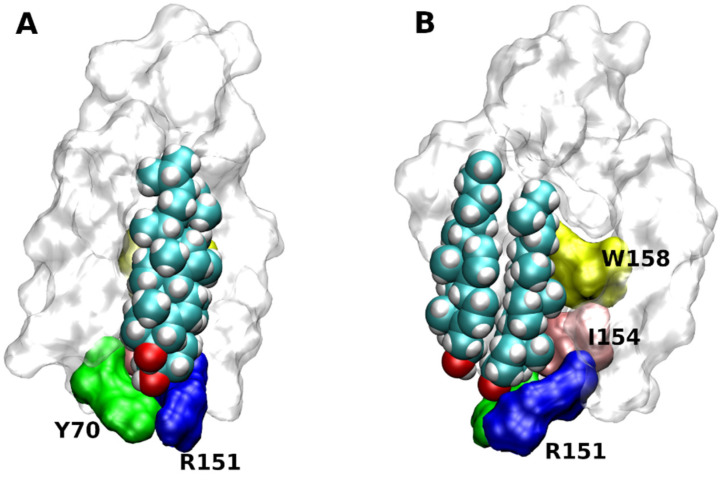
Structure of cholesterol consensus motif (CCM). CCM in the structure of β_2_-adrenergic receptor (3D4S) as viewed with TM2 (**A**) or TM4 (**B**) in front. Orientation, extracellular side up. Principal residues of CCM are coloured. Green—Y70 in TM2, blue R151, yellow—W158 and pink—I154 in TM4.

**Figure 5 ijms-22-01953-f005:**
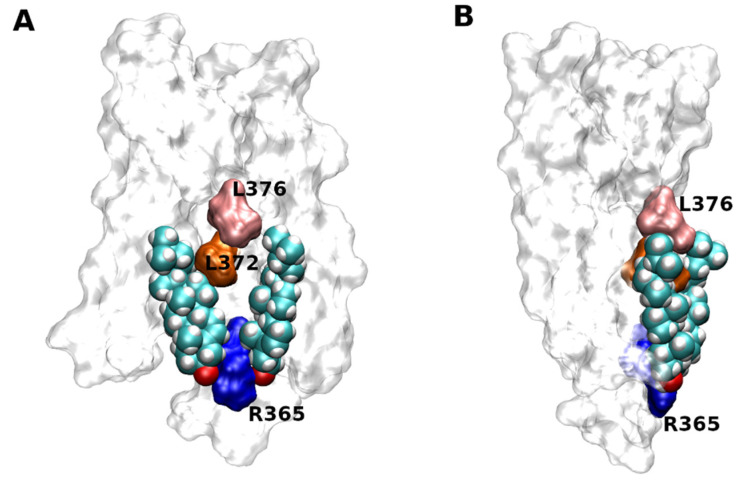
Docking of CLR to 5CXV structure of M_1_ muscarinic receptor. Two molecules of CLR docked to the structure of M_1_ muscarinic receptor (5CXV) as viewed with TM6 (**A**) or TM4 (**B**) in front. Orientation, extracellular side up. Principal residues in TM6 interacting with CLR are coloured. Blue—R365, orange—L372, pink—L376.

**Figure 6 ijms-22-01953-f006:**
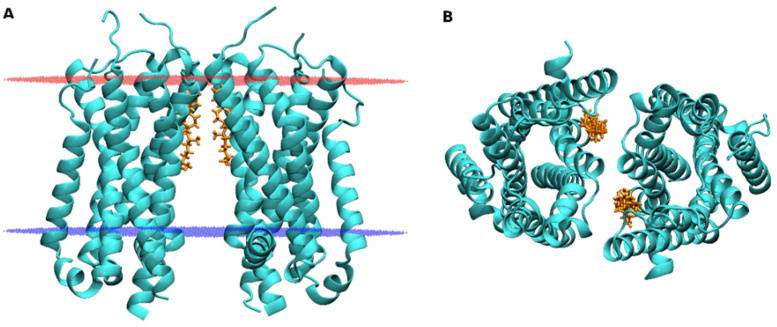
Dimer of the α_2C_-adrenergic receptor. The structure of α_2C_-adrenergic receptor dimer (6KUW) as viewed from the membrane side (**A**) and extracellular side (**B**). Blue—intracellular edge of the membrane; Red—extracellular edge of the membrane; Cyan—receptor; Gold—cholesterol.

**Figure 7 ijms-22-01953-f007:**
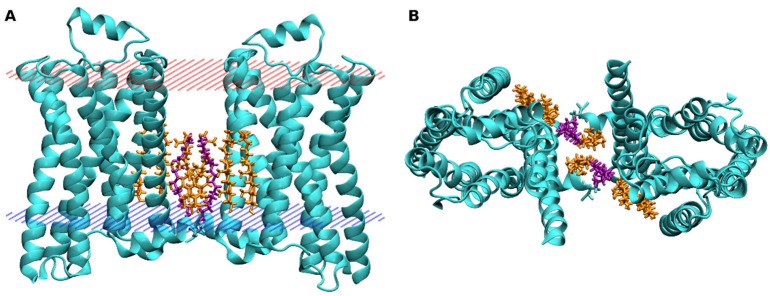
Model of the β_2_-adrenergic receptor dimer. Model of a dimer of β_2_-adrenergic receptor based on the crystal structure 2RH1 [16] as viewed from the membrane side (**A**) and extracellular side (**B**). Cyan—receptor; Gold—cholesterol; Purple—palmitic acid; Red—extracellular edge of the membrane; Blue—intracellular edge of the membrane.

**Table 1 ijms-22-01953-t001:** Cholesterol in crystal and cryo-EM structures. List of X-ray and cryo-EM (blue PDB codes) structures of Class A GPCRs containing cholesterol. Rec.—receptor subtype, Code—PDB ID code, G-prot.—a subclass of G-proteins mediating the primary response, Conf.—active or inactive conformation of the receptor, CLR—monomeric or dimeric state of CLR, Leaflet—location of CLR in the inner or outer leaflet of the membrane, TM—transmembrane helices CLR is interacting with, Ref.—reference.

Rec.	Code	G-prot.	Conf.	CLR	Leaflet	TM	Notes	Ref.
α_2C_	6KUW	G_i_	Inactive	Monomer	Out	1, 7	cholesterol is a part of the protomer-protomer interface	TBP
β_2_	2RH1	G_s_	Inactive	Dimer	In	2, 3, 4		[16]
				Monomer	In	1, 8		
β_2_	3D4S	G_s_	Inactive	Dimer	In	2, 3, 4		[17]
β_2_	6PS0	G_s_	Inactive	Monomer	In	2, 3, 4	6PS2, 6PS3, 6PS4, 6PS5, 6PS7 same	[18]
κOR	6PT2	G_i_	Active	Monomer	Out	6	only with one protomer	[19]
κOR	6B73	G_i_	Active	Monomer	Out	6		[20]
κOR	6VI4	G_i_	Inactive	Monomer	In	4, 5	only with one protomer	[20]
μOR	4DKL	G_i_	Inactive	Monomer	Out	6		[21]
μOR	5C1M	G_i_	Active	Monomer	Out	6		[22]
5-HT_2B_	4IB4	G_q_	Active	Monomer	In	1, 8		[23]
5-HT_2B_	5TVN	G_q_	Active	Monomer	In	1, 8		[24]
5-HT_2B_	6DRX	G_q_	Active	Monomer	In	1, 8	6DRY, 6DRZ, 6DS0 same	[25]
A_2A_	4EIY	G_s_	Inactive	Dimer	Out	6		[26]
				Monomer	Out	2, 3, 4		
A_2A_	5IU4	G_s_	Inactive	Dimer	Out	6	5UI7, 5UI8, 5UIA and 5UIB same	[27]
				Dimer	Out	2, 3, 4	5UIB only monomer.	
AT_1_	6OS1	G_q_	Active	Monomer	In	1, 8	6OS0 and 6OD1 no cholesterol	[28]
CB_1_	6N4B	G_i_	Active	Monomer	In	3, 4		[29]
				Monomer	In	3, 4	unexpected orientation with OH in the middle of the membrane	
CB_2_	6PT0	G_i_	Active	Dimer	Out	6		[30]
				Monomer	In	5, 6	parallel to membrane	
				Monomer	In	3, 4	parallel to membrane	
CCR9	5LWE		Inactive	Monomer	In	6		[31]
CLT2	6RZ7	G_q_	Inactive	Monomer	In	6	6RZ6, 6RZ9 dtto.	[32]
CXCR2	6LFM	G_i_	Active	Monomer	In	2, 3, 4	6LFO ditto, 6LFL no cholesterol	[33]
CXCR3	5WB2	G_i_	Active	Dimer	Out	6		[34]
ETB	5X93	G_q_	Inactive	Monomer	Out	1, 7	5XPR no cholesterol	[35]
FPR2	6LW5	G_i_	Active	Monomer	In	6		[36]
				Monomer	In	2, 3, 4		
FPR2	6OMM	G_i_	Active	Dimer	Out	1, 2		[37]
				Monomer	Out	6		
				Monomer	In	6		
				Monomer	In	3, 4, 5		
OTR	6TPK	G_q_	Inactive	Monomer	Out	4, 5		[38]
P2Y_12_	4NTJ	G_i_	Inactive	Monomer	Out	1, 7		[39]
				Monomer	In	3, 4	unexpected orientation, binding to Y in DRY motif	
P2Y_12_	4PXZ	G_i_	Inactive	Monomer	In	2, 3, 4		[39]
P2Y_1_	4XNV	G_q_	Inactive	Monomer	Out	2, 3, 4		[40]

**Table 2 ijms-22-01953-t002:** Cholesterol binding motifs and residues interacting with CLR. List of X-ray structures of Class A GPCRs containing cholesterol predicted CLR-binding motifs and CLR-interacting residues. Code—PDB ID code, CLR—monomeric or dimeric state of CLR, leaflet—location of CLR in the inner or outer leaflet of the membrane, TM—transmembrane helices CLR is interacting with.

Rec.	Code	CLR	Leaflet	TM	Predicted	Confirmed?	CLR-Interacting Residues
α_2C_	6KUW	Monomer	Out	1, 7	CCM	No	Q45, Y46, E112, K420
β_2_	2RH1	Dimer	In	2, 3, 4	CCM	Yes	Y70, T73, S74, R151, W158
		Monomer	In	1, 8			T56, C341_Plm
β_2_	3D4S	Dimer	In	2, 3, 4	CCM	Yes	Y70, T73, S74, R151, W158
β_2_	6PS0	Monomer	In	2, 3, 4	CCM	Yes	Y70, T73, S74, R151, W158, F166
κOR	6PT2	Monomer	Out	6	CCM	No	F280, D293 (o2)
κOR	6B73	Monomer	Out	6	CCM	No	T288, F293, T302, S303, S311
κOR	6VI4	Monomer	In	4, 5	CCM	No	F147, T150, Y157, H162
μOR	4DKL	Monomer	Out	6	CCM Q	Yes	T294, Y299, F313, Q314, S317
μOR	5C1M	Monomer	Out	6	CCM Q	Yes	T294, Y299, F313, Q314, S317
5-HT_2B_	4IB4	Monomer	In	1, 8	CCM	No	T73, Y394, Y399, C397_Plm
5-HT_2B_	5TVN	Monomer	In	1, 8	CCM	No	T73, Y394, Y399
5-HT_2B_	6DRX	Monomer	In	1, 8	CCM	No	T73, Y394, Y399
A_2A_	4EIY	Dimer	Out	6			F182, 183, 255, 258, S263, H264
		Monomer	Out	2, 3, 4	CCM	No	F70, F79, Q163 (o2)
A_2A_	5IU4	Dimer	Out	6			F182, 183, 255, 258, S263, H264
		Dimer	Out	2, 3, 4	CCM	No	F70, H75, F79, F133, Q163
AT_1_	6OS1	Monomer	In	1, 8	CCM	No	F39, F44, S47
CB_1_	6N4B	Monomer	In	3, 4	CRAC	Partly	F208, K232
		Monomer	In	3, 4			F208, Y215, H219, R220
CB_2_	6PT0	Dimer	Out	6	None	--	Q276, K279, F283
		Monomer	In	6			Y207, H211, W214, H217, R238, D240
		Monomer	In	3, 4			D130, Y137, T153, R149
CCR9	5LWE	Monomer	In	6	CCM	Yes	K254, T258, F263, F308, F319
CLT2	6RZ7	Monomer	In	5	CARC	Yes	S218, Y221, R226, F257
CXCR2	6LFM	Monomer	In	2, 3, 4	CRAC W	Yes	N89, N129, K163, W170, L174
CXCR3	5WB2	Dimer	Out	6	CCM	Yes	T247, F265, S268, R271, T276
ETB	5X93	Monomer	Out	1	None	--	Y102, T105
FPR2	6LW5	Monomer	In	6			S215
		Monomer	In	2, 3, 4	CRAC	No	N66, W150
FPR2	6OMM	Dimer	Out	1, 2		--	F37,
		Monomer	Out	6			F206, F255, W267
		Monomer	In	6			H229
		Monomer	In	3, 4, 5	CRAC	No	F118, H129, W132
OTR	6TPK	Monomer	Out	4, 5	CCM	No	F191, W195, Y200, W203
P2Y_12_	4NTJ	Monomer	Out	1, 7			F28, Y278, S282, W285
		Monomer	In	3, 4	CCM	No	Y123, Q124
P2Y_12_	4PXZ	Monomer	In	2, 3, 4	CCM	Yes	F51, S55, K64, N65, F106, K142, W149, F153
P2Y_1_	4XNV	Monomer	In	2, 3, 4	CCM	No	Y189, T221, Y217, S213

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
