# Peer review of "Allosteric Modulation of GPCRs of Class A by Cholesterol"

_ijms, 2021, doi:10.3390/ijms22041953_

Round 1
Reviewer 1 Report
In this manuscript, the authors provide a focussed review of the literature describing the class A GPCR and the regulation of its activity by cholesterol. Overall, the review is well written. I have the following suggestions for considerations by authors:
- A figure to show a schematic and structural representation of a representative GPCR highlighting various structural components of the receptor (such as described in the first paragraph of the Introduction section, the site of orthosteric ligand binding, and cholesterol binding motifs that are discussed later in the review) will be very helpful to the readers.
- In recent years, cryo-EM has been instrumental in determining three-dimensional structures of membrane proteins including GPCR. I recommend authors extend their analysis to include, if available, cryo-EM structures of cholesterol bound class A GPCRs.
- The authors describe an exciting perspective about the utility of cholesterol binding sites for designing drug or small molecule modulators for GPCRs. Are there specific examples of compounds that can bind to the cholesterol binding site to exert biological effects?
Author Response
We thank the reviewer for constructive comments.
1. We have added a schematic representation of GPCR of class A.
2. Cryo-EM structures are also included in the review. We amended the main text accordingly.
3. Neurosteroids have been shown to interact with many types of receptors, including serotonin, adrenergic and muscarinic. A possible therapeutic role in hypertension has been suggested for some neurosteroids. Neurosteroids have the same structural scaffold as cholesterol. Thus, they are likely to interact at the CLR-binding site.
Reviewer 2 Report
The manuscript by Jakubik and El-Fakahany reviews the specific receptor-binding sites of cholesterol (CLR) to rhodopsin-like GPCRs and changes in binding/signaling associated with different cholesterol levels. A large number of primary studies have addressed this issue; and both, structural and biological data support a modulating role of cholesterol for GPCR biology. The authors propose targeting the CLR binding sites as a novel and specific way to alter GPCR pharmacology. Overall, this is an interesting manuscript that aims at bringing together structural and functional observations on the interaction of cholesterol and GPCRs. However, a few specific points need to be addressed before the manuscript can be published (see below).
The first part concentrates on CLR binding sites that have been observed in crystallographic studies. It gives a good overview of observed binding sites, and tries to match them with consensus-motifs. It becomes clear that the known consensus motifs often have limited predictive value for the cholesterol binding site. This is, cholesterol binding might occur despite the absence of a consensus motif, or the observed binding site is different from the predicted site.
In the second part, biological/biochemical studies are reviewed that report on altered receptor function as a consequence of cholesterol interactions, with an emphasis on ‚receptor-specific‘ effects rather than indirect effects originating from altered membrane fluidity. Putative cholesterol-mediated effects on binding, signal transduction and dimerization are discussed in different receptor families, like the opioid or muscarinergic systems. Accordingly, CLR may function as positive or negative modulator of receptor affinity/activity. Unfortunately, there is little connection to the first (structural) part although crystal structures exist for many of these systems. In addition, the impact of cholesterol on receptor dimerization should be discussed with more caution.
Specific points:
First part:
- Different crystal structures are analyzed for the cholesterol binding sites. I miss a brief discussion as to how far crystallographic artefacts and crystallization conditions can contribute to (artificial) cholesterol binding sites.
- On page 7, line 204 and following, a few PDB accession numbers are listed. For clarity, the corresponding receptors should also be denoted in brackets.
Second part:
- For many receptors presented as examples, the authors miss the chance to link findings from the biological context to the crystallographically observed CLR binding site, and thus, speculation on the molecular mechanisms. For instance, at the oxytocin receptor, the location of CLR in the outer leaflet at TM4/5 connects with the observed positive effect on ligand binding affinity. In contrast, for the CB1 receptor, CLR binds in the inner leaflet according to the tables in part 1, which suggests a more complex interaction affecting ligand binding. For the muscarinergic receptors, CLR appears to affect ligand binding, but there was no CLR in the crystal structures according to Tables 1/2?
- The authors state CLR does not alter binding to the A2AR. However, the Selent lab has reported CLR dampens antagonist binding and regulates access of orthosteric ligands into the binding pocket (Nat Commun. 2017 Feb 21;8:14505.doi: 10.1038/ncomms14505), which should be included in the discussion of this receptor.
- Functional response of GPCRs: The relevance of the so-called ‘ionic lock’ in the sequence of receptor activation is emphasized. However, most GPCR do not have the corresponding residue at position 6.30 and the ionic lock rather seems to be an additional ‘safety-lock’ among aminergic and purinergic receptors (reviewed e.g. Tehan et al. Pharmacol Ther. 2014 Jul;143(1):51-60.; Zhou et al. Elife. 2019 Dec 19;8:e50279.) However, the conserved R3.50 is a direct contact site for the C terminus of the G alpha subunit, which can rationalize effects of CLR binding sites near this residue.
- Effects on dimerization: In particular the beginning of this section gives the impression that many rhodopsin-like GPCRs REQUIRE dimerization for their biological activity, and the presented literature examples appear biased in that sense. GPCR dimerization has been a very controversial field for many years, but the prevalent ‘consensus’ in light of all currently available data is that rhodopsin-like GPCR are functional as monomers (while class C GPCRs are obligate dimers, which are, however, not covered in this review), AND their activity can be ‘tuned’ by homo- and heterodimerization. The references should be selected and presented in a more balanced way to appreciate both aspects. Moreover, high protein concentration during biochemical/biophysical studies (overexpression, crystallography) can favor dimerization, which needs to be considered for interpretation of these studies (see comment below for opioid receptors; but also applies to the discussion of b2AR and 5-HT1a dimers on page 13, starting at line 369)
- In this regard, the dimerization of opioid receptors has been very thoroughly investigated after the paper of Jordan and Devi (1999). In fact, reference 75 (Kuszak et al., 2009), which is cited for confirmation of mu-OR dimerization by fluorescence techniques, actually emphasizes that the monomeric receptor suffices as functional unit. Moreover, later structural studies using crystallography or cryo-EM have found agonist-bound mu-OR monomers in complex with nanobody and Gi, respectively (Huang et al. Nature. 2015 Aug 20;524(7565):315-21.; Koehl et al. Nature. 2018 Jun;558(7711):547-552). In addition, unbiased MD simulations and FRAP-experiments found limited mu-opioid dimerization at physiological concentration (Meral et al. Sci Rep. 2018 May 16;8(1):7705.). However, a very recent single-molecule study pointed out that specific agonists can specifically enhance mu-OR dimerization, underlining the aspect of tuning opioid receptor activity by dimerization (Möller et al. Nat Chem Biol. 2020 Sep;16(9):946-954.).
- Along the same line, reference 78 (Bayburt et al., 2011), which is cited for a potential requirement of rhodopsin dimers, emphasizes that rhodopsin monomers are sufficient for arrestin recruitment. This needs to be amended.
Formal aspects/Language:
There are several formatting errors in the list of references (bold/non-bold; in ref. 60, the doi is replaced by keywords)
Page 3, l.87 …many of GPCRs.
Page 11, l.291-292, should read: …CLR pools may restrict the ability of the receptor to attain certain conformations.
Page 11, l.293, should read: …both mechanisms are important for the functional response of the receptor.
Author Response
We thank the reviewer for constructive comments and pointing out errors and typos.
First part
We have added a warning on possible crystallographic artefacts of CLR. The unnatural orientation of CLR molecules mentioned in the text and Table 1 provide a piece of direct evidence for the artefacts.
On page 7, we have added receptor names to corresponding PDB IDs.
Second part
We thank the reviewer for the helpful suggestion. We have added references to structural data in the second part of our manuscript.
Regarding CLR effects at muscarinic receptors: Yes, there is ample evidence that membrane CLR affects binding and activation of muscarinic receptors. Interaction of CLR with R6.35 was proposed by site-directed mutagenesis. However, CLR was not found co-crystallized at any of 16 published structures. Only M1 structures 5CXV and 6WJC contain cholesteryl hemisuccinate (CHS). We do not overview CHS “binding” in the manuscript as it may be the artefact of the solubilization process where CHS is used to solubilize membranes.
Ad Guixà-González et al. Membrane cholesterol access into a G-protein-coupled receptor, Nat. Commun. 8 (2017). https://doi.org/10.1038/ncomms14505. We have added a comment that water-soluble cholesterol (WSC) at millimolar concentrations may enter the binding cavity of the A2A receptor and competitively inhibit [3H]ZM241385 binding. However, as we state in the manuscript, membrane cholesterol displays nanomolar affinity. Moreover, in the mentioned paper no complete inhibition was reached even by 3 mM WSC (Figure 4) and cholesterol entered the orthosteric binding cavity only in one out of four MD replicas (Figure 5).
We have amended the paragraph on receptor activation by stating that R3.50 in an inactive conformation forms ionic lock with D or E3.49, or in some cases with E6.30, that prevents it from interaction with the C-terminus of the G‑protein α‑subunit.
Our review summarizes allosteric modulation of GPCRs of class A by CLR. Naturally, we focus on the role of CLR in receptor oligomerization. Thus, it may seem biased towards GPCR oligomerization. To dispel such misconception, we have amended the opening paragraph of the section. We acknowledge that the vast majority of GPCRs of class A are fully functional as monomers. The discussion on the precise mechanism of GPCR oligomerization is tangential to the topic of the review. Thus we have made it as short as possible.
We have amended the paragraph on dimerization of opioid receptors. Importantly, although, monomers of opioid receptors are functional, the changes in receptor pharmacology upon co-expression of μ- and δ‑opioid receptors strongly indicate functional oligomerization of opioid receptors that is important from the pharmacological point of view.
We apologize for erroneous reference to Bayburt et al., 2011. We have rewritten the paragraph.
We thank the reviewer for pointing out typos. We have corrected them.
Round 2
Reviewer 1 Report
I am overall happy with the revised manuscript. A minor note: Table 1 is titled "Cholesterol in crystal structures" but it includes (though only a few) cryo-EM structures e.g., 6LFM and 6LFO. It is confusing as I see that cryo-EM structures were also included in the original manuscript. Please revise this as appropriate.
Author Response
We have amended the title of Table 1 and changed PDB codes of cryo-EM structures to blue for better orientation.
Reviewer 2 Report
All of my concerns have been addressed and the mansucript has been substantially improved. Possible correlations of structural and functional studies are now clear; and the section on GPCR dimerization integrates and balances different studies.
Author Response
Thank you once again, for your constructive attitude.